# Storage Time as an Index for Varietal Prediction of Mango Ripening: A Systemic Approach Validated on Five Senegalese Varieties

**DOI:** 10.3390/foods11233759

**Published:** 2022-11-22

**Authors:** Mor Dieye, Nafissatou Diop Ndiaye, Joseph Bassama, Christian Mertz, Christophe Bugaud, Paterne Diatta, Mady Cissé

**Affiliations:** 1Institut de Technologie Alimentaire (ITA) Route des Pères Maristes, Hann Bel Air, Dakar BP 2765, Senegal; 2Laboratoire d’Electrochimie et des Procédés Membranaires, Ecole Supérieure Polytechnique, Université Cheikh Anta Diop, Dakar BP 5005, Senegal; 3Faculté des Sciences Agronomiques, Aquaculture et Technologie Alimentaire, Université Gaston Berger de Saint-Louis, Route de Ngallèle, Saint-Louis BP 234, Senegal; 4CIRAD, UMR Qualisud, 34398 Montpellier, France; 5Qualisud, Univ Montpellier, Institut Agro, CIRAD, Avignon Université, Université de la Réunion, 34398 Montpellier, France; 6Centre de Recherches Agricoles (CRA) de Djibélor (Ziguinchor), Institut Sénégalais de Recherches Agricoles (ISRA), Hann Bel Air, Route des Hydrocarbures, Dakar BP 3120, Senegal

**Keywords:** ripening storage time, Senegalese mango varieties, varietal prediction, multi-linear regression, Boukodiekhal, Diourou, Papaye, Sierra Leone, Sewe

## Abstract

*Mangifera indica* species presents a wide varietal diversity in terms of fruit size and morphology and also of physicochemical and organoleptic properties of the pulp. In Senegal, in addition to the well-known export varieties, such as ‘Kent’, local varieties have been little studied particularly during ripening. This study aims to propose prediction models integrating variables deduced from varietal characteristics. Five mango varieties (‘Diourou’, ‘Papaye’, ‘Sierraleone’, ‘Boukodiekhal’ and ‘Sewe’) endemic to Senegal were characterized at harvest and followed during ripening storage. Caliber parameters were determined at green-mature stage as well as storage (25 °C) weight losses. Considering the ‘ripening storage time’ (R_ST_) variable as ripeness level index, intra-varietal prediction models were built by multi-linear regression (R^2^ = 0.98) using pulp pH, soluble solid content (SSC) and Hue angle. In addition to these physicochemical parameters, variety-specific size, shape and weight loss parameters, were additional variables in multi-linear models (R^2^ = 0.97) for multi-varietal prediction of R_ST_. Results showed that storage time, which was the most influential factor on the pH, SSC and Hue, can be used as a response for varietal prediction of mango ripening. As a decision support tool, theses statistical models, validated on two seasons, will contribute to reduce post-harvest losses and enhance mango value chain through a better ripening process monitoring.

## 1. Introduction

Mango (*Mangifera indica* L.) is one of the most consumed fruits in the world, Africa being the region of the highest increase in production in the last decade [1]. There are more than a thousand varieties of mango, with diversity in shape, size, color, texture, and nutritional properties [2]. In Senegal, mango production increased by 30% between 2009 and 2019, peaking at 132,000 tons in 2017 [3]. Senegalese mangoes have covered, on average over the last 5 years, 24% of EU imports [4], providing a large size; pulpy and not too fibrous fruits mainly of ‘Kent’ and ‘Keitt’ varieties. However, in addition to these so-called ‘export’ varieties, there are approximately 30 local varieties [5] that can be developed. Some of these varieties are characterized by small fruit size (<200 g), fibrous pulp, but pleasant and interesting flavors [6]. The most endemic and marketed locally and sub-regionally varieties are, in decreasing order of farm area: ‘Diourou’; ‘Boukodiékhal’; ‘Sierra Léone’; ‘Sewe’ and ‘Papaye’ [7]. An optimal exploitation of these varieties would reduce losses estimated at approximately 30–50% of production in the Niayes zone and even 60% in the South zone [5,8]. In these main production zones, deficits in storage infrastructure, in addition to low processing level (1 to 3%), constitute a weak point in the value chain [7]. Enhancing the nutritional and economic potential of these local varieties would require more information on their characteristics and technological abilities, particularly for ripening storage. Harvested at the green-mature stage, control of this stage will make it possible to anticipate attacks by fruit flies, the main cause of losses. On the other hand, it will guarantee a better controlled ripening for fresh consumption, and the quality of the raw material for processing. Beyond this local context, studies on physiological and biochemical aspects have helped to understand the variability of behavior between different varieties of mangoes during ripening. Translated into experimental parameters, this varietal variability could be taken into account in the monitoring and prediction of mango quality during ripening storage.

Local mango varieties in Senegal have been the subject of a few agronomic [1], biomedical [9] and biochemical [10] studies. However, there is a lack of data on the size properties and physicochemical characteristics of these poorly known varieties. Data, such as mass, dry matter, soluble dry extract, and coloration (pulp and skin) of these mangoes, would be good indicators of the quality and technological potential of these varieties. Used by pickers to decide the ideal harvest stage, size and mass are size parameters, indicators of the physiological maturity of the fruit in pre-harvest [11]. 3D parameters increase during mango growth and determine its morphology at the mature stage. They are strongly correlated (R^2^ = 0.97) with mango mass, following a variety-specific equation [12]. Therefore, the experimental acquisition of these size parameters would allow the generation of exploitable varietal parameters.

In climacteric fruits, the metabolic changes of ripening are related to respiratory activity and transpiratory loss in water mass, resulting in a decrease in fruit weight [13] and wilting [14]. As function of storage temperature, these two fundamental aspects determine the fruit shelf life and quality at ripening storage [15]. Water loss depends on the water vapor pressure at the product surface and also on the cuticle conductance [16]. However, as shown by [17], anatomical differences between mango varieties are reflected at the cellular level, on the resistance to diffusion [18] through the walls, especially from the interior of the fruit to the ambient atmosphere [19]. From then on, exploitable varietal parameters could be deduced from experimental mango weight loss kinetics.

Physiological and biochemical changes in mango ripening correspond to changes in physicochemical parameters. In different studies, the monitoring of these parameters has allowed a prediction of the ripening stage, which is essential for the post-harvest treatment of climacteric fruits. An index to predict ripeness level would allow post-harvest operators, processors and researchers to monitor and control fruit quality in industry. As a function of evolving parameters during ripening, indices, such as the ripening index (RPI) [20,21,22,23], ripening index (Im) [24] or the ripening class index (RCI) [25], have been studied. Nambi et al. [26] proposed the ripening index (I_R_) quantifying the ripeness level evolution, declined in physicochemical parameters of the pulp; texture and colors of mango. However, the prediction parameters of this IR, reflected on ‘Alphonso’ and ‘Banganapali’, varietal ripening specificities [27] involving anatomical differences that, according to Paul et al. [17], impact the respiratory intensity of mango fruits. These differences induce as many equations as varieties for the same prediction model. Moreover, it appears from this study that the same level of ripening (same I_R_) can correspond to different ranges of parameters (Acidity, SSC, Color …) from one variety to another. At this knowledge state, a way to integrate into the models, parameters reflecting varietal variability, would fall under a consistent systemic approach. In this case, the duration of storage, a quantitative variable specifically chosen with a scale of experimental values, would be interesting as an index to predict mango ripening.

The purpose by this study approach is to build prediction models of mango ripening, integrating the ‘variety’ factor as varietal variable parameters. These varietal-specific parameters will be deduced from the characterization of five contrasting varieties. That will also improve knowledge on technological (ripening/processing) aptitude of the ‘Diourou’, ‘Papaye’, ‘Sierra Léone’, ‘Boukodiekhal’ and ‘Sewe’ mangoes. These Senegalese varieties, still under-exploited, are all or partially cited by Ndimanya et Strebelle [28] and/or Belmin [29].

## 2. Materials and Methods

### 2.1. Plant Material

The study focused on five local varieties: ‘Diourou’, ‘Papaye’, ‘Sierra Léone’ from the South zone (SZ); and ‘Boukodiekhal’, ‘Sewe’ from the Niayes zone (NZ) of Senegal. In the NZ, the experimental orchard is located in the Monastery farm in the commune of Keur Moussa—Thiès region (14°46′49.5188″ N, 17°6′54.8028″ W). In the SZ, a family orchard in the commune of Niaguiss—Ziguinchor region (12°32′23.9118″ N, 16°12′27.1235″ W) was used as an experimental orchard, in collaboration with Agricultural Research Center (CRA Djibélor) of the Senegalese Institute of Agricultural Research (ISRA). The study was carried out over two harvesting campaigns following two consecutive seasons.

### 2.2. Sampling

Harvesting was done supervised by an experienced picker, according to visual criteria (color, size, appearance of the stalk, etc.).

#### 2.2.1. Batch for Varietal Characterization and Model Calibration (Campaign 1, 2019 Season)

For each variety, 125 mangoes were harvested at the green-mature stage. For each variety, 40 mangoes were used for measurements of caliber parameters (weight and size) at harvest stage.

#### 2.2.2. Batch for Model Validation (Campaign 2, 2020 Season)

These sample lots consisted of the same varieties harvested from the same respective sites. For each variety, a sample of 80 mangoes were used. Divided into three parts, each sample was composed of unripe-mature, early ripe and ripe mangoes.

### 2.3. Mango Caliber at Harvest

Fruit weight at harvest was determined by direct weighing on a 0.01 g precision scale. The maximum length, width and thickness of L_max_, l_max_ and e_max_ (Figure 1) of the fruit, were measured with digital calipers at 0.001 mm precision scale.

### 2.4. Ripening Storage

Mangoes were then stored under controlled conditions (CC) in a dedicated room at 25 °C/86% relative humidity (RH), at the Fruit and Vegetables research unit of the Food Technology Institute (ITA). A ripening storage under ambient conditions (CA) at 29 °C/57% RH was done in parallel to serve as a control.

Harvested in the South zone, at the green-mature stage, then stored under the same experimental conditions, a batch of ‘Kent’ mangoes, the most exported variety and the most processed locally, was used for characteristic references.

### 2.5. Monitoring of Physicochemical Parameters during Ripening

Dry matter (DM), titratable acidity (TA), pH, color and soluble solids content (SSC) of mango pulp were measured at harvest (green-mature stage) and then monitored during the ripening storage. Five fruits were sampled after 4, 8 and 12 days of storage. Each mango was peeled and the pulp was ground in a blender. The resulting puree was used directly for measurements of dry matter, pH, titratable acidity, soluble solid contents and color, while the remainder was stored at −18 °C for further analysis. SSC of mango pulp was measured by digital refractometer (ATAGO^®^ PAL-3, Atago Instruments, Tokyo, Japan). The pH and TA measurements of the pulp were performed with a titrator (Titroline 96, Schott-Geräte GmbH, Hofheim am Taunus, Germany). The pH was read directly, while a 0.1 N sodium hydroxide solution (Sigma-Aldrich, St. Louis, MO, USA) was used for the determination of TA expressed in g ± 0.01 of citric acid (the major organic acid in mango [30]) per 100 g of puree. DM was determined according to AOAC method 934.06/37.1.10 and expressed as g ± 0.01 of dry matter per 100 g of pulp. Color measurements were performed using a chromameter (Minolta CR-400, Tokyo, Japan) on the well homogenized puree. The parameters of the CIE color space L*, a*, b* are: luminance L* determined on a scale from black to white; a* from green to red and b* from blue to yellow. The color space L*, C* (Chroma) and H* (Hue angle) allowed to exploit the color parameters. To follow the evolution of the pulp color, the value of the Hue angle is obtained following the Formula (1):Hue = tan^−1^ (b*/a*)(1)

Furthermore, these different parameters can be expressed as relative values of a variable in a α_p_ form (‘_p_’ denoting the parameter pH, TA, SSC or Hue considered). The αp value was obtained for each parameter, by dividing its experimental value X_p_(t) at storage time t, by its initial value X_p_(0). For TA and SSC, this resulted in dimensionless variables calculated by the following formula:α_p_ = X_p_(t)/X_p_(0)(2)

For each parameter ‘_p_’ considered, X_p_(0) is the average calculated on *n* = 5 fruits (×2 measurements per fruit) of the experimental values at t = 0 (mango stage at the beginning of storage).

### 2.6. Monitoring of Weight Losses during Ripening Storage

For each variety, weight losses in CC and CA storage, were followed by weighing a same fruit, at harvest, then at 4, 8 and 12 days of ripening storage. For each variety, the percentage of weight loss, obtained by Formula 3, was calculated as an average value on five fruits (*n* = 5).
Weight loss (%) = [(M(0) − M(t))/M(0)] × 100(3)

M(0) is the initial weight (green-mature) of the mango and M(t) the weight at a storage time t (days).

### 2.7. Statistical Analyses

All statistical analyses were carried out using XLSTAT software (version 2022.1.1, Addinsoft, 75018 Paris, France). Descriptive analyses (means, standard deviations, coefficient of variation) were done with the measurement trials the means on each mango fruit. Analyses of variance (ANOVA) on five replicates (five fruit) and a Tukey test (*p* < 0.05) allowed for comparison between varieties and/or storage time. Simple linear regression models were used to monitor storage weight losses. Significant singular and interacting effects of the factors ‘variety’ and ‘storage time’ were first identified by analysis of covariance (ANCOVA). After verification of normal distribution of the residuals (*p*-value > 0.05; Shapiro–Wilk test), the multi-linear regressions were tested for the storage time prediction. For intra-varietal prediction, models have been calibrated (*n* = 20 fruit) at campaign 1 and then cross-validated (*n* = 15 fruit) at campaign 2. At the same campaigns, multi-varietal models have been calibrated (*n* = 100 fruit) and then validated (*n* = 75 fruit) for statistical prediction by multi-linear regression (MLR). As a ripening stage index, the ripening storage time R_ST_ was a response variable while physicochemical and varietal parameters constitute explanatory variables. The performance of the models was evaluated by the coefficient of determination (R^2^) and the root mean square error (RMSE), calculated with the following formula:(4)$$\mathrm{RMSE}=\sqrt{\;\frac{1}{n-f}\;\sum_{i=1}^{n}{\left(R_{\mathrm{ST}}-R_{\mathrm{ST}.\mathrm{est}}\right)}^{2}}$$

R_ST_ and R_ST.est_ are, respectively, experimental and estimated values of the response variable; *n* is the number of observations (number of mangoes), while f is the number of coefficients in the linear equation.

Since the variable ‘ripening storage time’ spans a wide range (0 to 12 days) with different mean values, e.g., fruits at different ripening stages, the standard error with respect to the measured value [12,31] proved useful. Thus, the mean relative deviation (MRD) was calculated according to the following formula:(5)$$\mathrm{MRD}=\sqrt{\;\frac{1}{n-f}\;\sum_{i=1}^{n}(\frac{R_{\mathrm{ST}}-R_{\mathrm{ST}.\mathrm{est}}}{R_{\mathrm{ST}}}{)}^{2}}$$

For model comparison, the Akaike Information Criterion (AIC) was used. The lower the AIC of the model, the better the compromise between a good response prediction and an optimal coefficients number in the equation [32].

### 2.8. Concept of Ripening Prediction

SSC, pH and Hue were the three physicochemical variables selected as quality parameters of ripened mango. Storage time (R_ST_), the most influential variable [33] on the evolution of physicochemical parameters (according to ANCOVAs), was defined as an index of ripening stage.

#### 2.8.1. Intra-Varietal Prediction

The significant effect of the ‘variety’ factor suggests in a first step a singularized approach leading to a statistical model for each variety. The ripening storage time (R_ST_) is thus predicted as a response variable, explained by the pH, SSC and Hue variables on MLR. The ripening index R_ST_ becomes a function defined by the following equation:f (SSC, pH, Hue) = R_ST_(6)

#### 2.8.2. Multi-Varietal Prediction: Definition of Varietal Parameters as Varietal Variables

The influence of ‘variety’ in interaction with ‘storage time’ suggests a systemic approach to prediction. Thus, the challenge lies in the transcription of the qualitative variable ‘variety’ into quantitative varietal variables, exploitable in MLR. On the basis of the characterization data at harvest and the follow-up of the physicochemical parameters during ripening storage, variety-specific parameters were provided as varietal variables.

Let C_f_ (coefficient of form) be a parameter based on the three dimensions of mango form. ANOVAs with difference tests (Tukey’s Honest Significant Difference test, *p* < 0.05) were used to compare the different varieties and to determine the specific mean value for each variety. C_f_ is calculated by the following equation:(7)$$C_{f}=\frac{L_{\max}\times l_{\max}}{e_{\max}}$$

Let C_c_ (coefficient of caliber) be the linear regression slope between fruit weight and the product of the three dimensions (L × l × e). As highlighted in perspective in the work of Spreer and Müller [12], this variety-specific cefficient can be considered as a varietal parameter, illustrative of the characteristic size of a variety.

Let C_tr_ (coefficient of transpiration) be the linear regression slope of mango weight loss versus storage time. Representing the weight loss rate, this coefficient is considered here as a physiological varietal parameter, illustrating the variety-specific weight loss by transpiration.

The parameters of form (C_f_), size (C_c_) and transpiration (C_tr_) are thus varietal factors that can be used as quantitative explanatory variables in a multivariate model. In this approach, the R_ST_ index will be a function defined by the following equation:f (C_f_, C_c_, C_tr_, pH, SSC, Hue) = R_ST_(8)

The response variable R_ST_ was predicted by six explanatory variables (pH, SSC, Hue, C_tr_, C_f_ and C_c_) on MLR model, calibrated (100 mangoes) and cross-validated (75 mangoes) on two different seasons. The R_ST_ prediction model using only pH, SSC and Hue as explanatory variables will be used as a control to compare and observe at the same time the effect of varietal parameters on the accuracy of the prediction models.

Variables in α_p_ format, allowed models to take into account the advancement of ripeness in each parameter expressed in relative value.

## 3. Results and Discussion

### 3.1. Caliber Parameters and Shape of Mangoes at Green-Mature Stage

Fruit weight and 3D parameters (L_max_, l_max_, e_max_) of the different varieties were assessed at green-mature stage (Table 1). The C_f_ parameters calculated, characteristic of the geometry of the variety as well as its form, are also presented in this table.

Significant differences were found overall between the local varieties. Weights vary from the smallest ‘Sewe’ mangoes (102.5 g), the smaller ‘Sierra Leone’ (214.4 g), the larger and almost identical ‘Boukodiekhal’ (469.7 g), ‘Diourou’ (442.2 g) and ‘Papaye’ (423.3 g). According to the codex standard (STAN 184-1993), these varieties can be classified according to size codes, in category A (200 to 350 g), B (351 to 550 g) or in C (551 to 800 g) with ‘Kent’ mangoes (600.65 g). Between the different varieties, the mango parameters show a relative caliber homogeneity into the harvest lot. This is reflected in low average coefficients of variation (13%, 1%, 7%, 6% and 8%), respectively, on the weight, length, width and thickness of the fruit. Concerning the fruit weight at harvest, the coefficients of variation at 9% for ‘Boukodiekhal’, 12% for ‘Diourou’ and ‘Papaye’, and to 14% for ‘Sierra Léone’ testify to a good control of the harvesting stage to guarantee a homogeneous maturity of the sample lot. On the other hand, higher variability (23%) noticed on ‘Sewe’ variety can be explained by the difficulty to select mature its small and grappled fruits. In addition to facilitating the handling and fruit transportation, the uniformity of the batches is a factor of attraction for the consumer. In agro-industry, fruit size is a key parameter for sizing of sorting/washing or pitting machines, optimal storage methods, flow management, etc.

Concerning the 3D parameters, there are similarities in L_max_ (‘Boukodiekhal’ vs. ‘Papaye’), l_max_ (‘Papaye’ vs. ‘Sierra Léone’) and e_max_ (‘Boukodiekhal’ vs. ‘Papaye’). However, results revealed that two varieties with different L_max_ and l_max_ (e.g., ‘Papaye’ vs. ‘Sierra Léone’) can be similar in e_max_. Since dimensions taken individually are very contrasted, it would be relevant to define a composite size parameter that would be quite edifying on the geometry of the fruit. The ‘Boukodiekhal’ variety, for example, is distinguished from the ‘Papaye’ by its larger l_max_, which gives it an elliptical shape. This morphological contrast is thus translated by a difference of (L × e)/l value noted C*_f_* (Table 1). By extending to the other varieties and comparing also (L × l)/e and (l × e)/L values, the lateral axis surface (L × l) related to the thickness (e), proved to be the only discriminant for all the varieties. Each variety would thus have a C_f_ mean value (*n* = 40), characteristic of its shape at mature stage. Therefore, elliptical shapes can be distinguished in the varieties ‘Boukodiekhal’, ‘Diourou’ and ‘Sierra Léone’, in contrast to the oblong shapes of ‘Papaye’ and ‘Sewe’. Information on shape in three 3D parameters is valuable for the evaluation of fruit quality. Caliber parameters of these different varieties will help to enrich databases for calibration of rapid measuring devices. Kabutey et al. [34] proposed 3D virtual models of mangoes, by an Intel RealSense 3D scanner while morphological information on pear apple were collected by Wang and Chen [35], using a Kinect depth camera. In pre-harvest, the RGB-D images method, to estimate mango size on tree would be convenient and fast to implement but cannot be used under intense sunlight [11]. Furthermore, there is a variety-specific linear relationship between equivalent polyhedral volume V (L_max_ × l_max_ × e_max_) and the fruit weight [12]. The linear regression (y = C_c_ x) slopes shown in Figure 2 are variety-specific parameters. These parameters will be used as varietal variables called here caliber parameters (C_c_).

### 3.2. Weight Losses during Mango Ripening Storage

Evolution of weight losses has been measured on days 4, 8 and 12 of storage, under controlled conditions (CC) and in parallel, under ambient conditions (CA). The losses are expressed as a percentage of the initial (day 0) fruit weight. In all samples, a linear (R^2^ = 0.9) decrease in weight was observed throughout storage.

In CA, loss rates (Table 2) are significantly higher at all measurement points compared to CC, where the observed losses are slower or even insignificant. In both CA and CC, ‘Boukodiekhal’ shows the more important loss percentages. All varieties taken together, the maximum percentage losses recorded at 12 ripening storage days was 12.2% and 27.2%, respectively, in CC and CA. This difference could be explained by the relative humidity which is lower in ambient conditions, thus favoring the loss in weight by physiological transpiration of these climacteric fruits. The ‘Boukodiekhal’, ‘Sierra Leone’ and ‘Sewe’ mangoes show on average the highest rate in CA, losing 27.2%, 25.5% and 23% of their initial weight, respectively. This is followed by ‘Diourou’ (16.7%), which was slightly more sensitive to storage, compared to ‘Papaye’ (13.2%) (Table 2). These observations provide nuances to the finding of Rathore et al. [36]. The latter, referring to Doreyappa et al. [37], mentioned an influence of fruit size on weight loss rate. After 12 days ripening, Table 2 shows significant differences and similarities between mangoes of contrasted sizes (‘Kent’ vs. ‘Papaye’ or ‘Boukodiekhal’ vs. ‘Sewe’).

For CC storage, Table 2 report less drastic loss rates for all varieties studied. Nevertheless, compared to the reference ‘Kent’ (4.7%), ‘Papaye’ (6.7%) is the least sensitive to losses compared to other local varieties, which lost on average 9.4% of their initial weight. Controlled condition (CC) ripening induced lower weight losses, corresponding to lower regression slopes (Table 2) and better linear fit (higher R^2^ values) to all varieties. Responsible for 92% to 97% of tomato weight loss [38], transpiration is the major pathway of physiological dehydration in fruits and vegetables [39]. Calculated loss levels in CC are similar to those reported in different works.

Under the same temperature condition (25 °C), An et Paull [40] reported 7.9% loss in weight on ‘Sunset’ papaya after 10 days of storage. Under ripening at 25 °C/60–70% RH, Okoth et al. [41] reported weight losses approximately 7.1% and 4.9% at 6–7 days on ‘Apple’ and ‘Ngowe’ mangoes, respectively, and 10.1% at 9–10 days for ‘Kent’. Controlled ripening storage involves temperature and humidity conditions. Packaging in a controlled or modified atmosphere makes it possible to prolong the post-harvest conservation of mangoes, particularly by reducing weight losses [42]. These weight losses affect the quality rating (loss of firmness, color rating, wilting) of fruits and vegetables [43].

Mango wilting, observed earlier in CA than in CC storage, results from significant water loss by transpiration. The results in Table 2 show that for this study that ripening in CC (25 °C/86% HR) reduces weight losses by 50% on average, compared to AC. These results further highlight a significant effect of ripening storage condition on mango fruit behavior.

As observed by Perez et al. [44], on avocado fruit, the increase in weight loss follows a linear regression whatever the storage conditions. Transpiration water loss is mainly by diffusion through the cuticle [45,46], whose barrier properties [47] may vary according to varietal anatomical characteristics, in particular cell arrangement patterns and intercellular spaces according to Paul et al. [17]. The regression slope (C_tr_) in CC (Table 2), thus reflecting a weight loss rate, is variety-specific. It represents in the following the transpiration parameter defined as a varietal variable in regression models.

### 3.3. Physicochemical Parameters of Mango Pulp at Harvest and Ripening Storage

Changes in soluble solid content (SSC), dry matter (DM), pH, titratable acidity (TA) and color (Hue) pulp during ripening can be observed in Figure 3a–e.

#### 3.3.1. Dry Matter Contents Indicated Physiological Maturity

From harvest stage to ripeness, no significant changes in dry matter content were observed in all the varieties studied. On average for the five local varieties, dry matter contents varied from 15.75% at harvest to 16.22% at 12 days of ripening storage. Dry matter contents measured at the green-mature stage ranged from 15.27% for ‘Sierra Leone’ to 16.45% for ‘Boukodiekhal’. These values, corresponding to the physiological maturity of ‘Osteen’ mangoes harvested at 133–140 days after full bloom [48], indicate a satisfactory maturity. In fact, when mango fruit reaches 14% dry matter, it is considered mature enough to be picked and ripen properly with good flavor [49]. In a previous study, ‘Mahajanaka’ mangoes of excellent taste quality had higher DM contents at harvest [50]. The level of dry matter content is a good indicator of maturity for harvest time [51,52]. The increase in DM during fruit maturing, implies an accumulation of organic substances necessary for the ripening process. For the remainder of the study, the low variability of these dry matter contents (Figure 3a), reflects an intra and inter-variety homogeneity of the sample lots maturity, which ensures reliability of subsequent comparisons.

#### 3.3.2. Soluble Solids Content (SSC), Acidity (AT, pH) and Color (Hue) of Mango Pulp at Harvest (Mature-Green) Stage

SSC at harvest shown less contrast comparing varieties. The maximum SSC were 9.35 ± 0.85% and 8.93 ± 0.55% for ‘Sierra Leone’ and ‘Diourou’, respectively. The varieties ‘Boukodiekhal’, ‘Papaye’ and ‘Sewe’, not significantly different, have an average SSC of approximately 7.51%. Comparing the Hue angle values at the mature-green stage, ‘Papaye’, ‘Sierra Léone’ and, to a lesser extent, ‘Sewe’ mangoes, presented a light greenish pulp color, close to that of ‘Kent’. Without significant difference, ‘Diourou’ and ‘Boukodiekhal’ had maximum Hue angles of 100.3 and 99.8, respectively. Concerning titratable acidity, at mature-green stage, the ‘Papaye’ variety at 4.09 ± 0.31 g/100 g and ‘Diourou’ at 3.24 ± 0.30 g/100 g have shown the highest levels compared to the average of 1.19 ± 0.12 g/100 g of the ‘Sewe’ and ‘Sierra Léone’ varieties. This acidity is mainly due to the organic acid composition of the pulp. These compounds are necessary for aerobic metabolism and are aromatic constituents that contribute to fruit quality and organoleptic properties [53].

#### 3.3.3. Soluble Solids Content (SSC), Acidity (AT, pH) and Color (Hue) of Mango Pulp during Ripening Storage

SSC increased significantly from 4 days of storage, then stabilized at 15.55 ± 0.47 °Bx and 14.51 ± 0.85 °Bx, respectively, for ‘Diourou’ and ‘Papaye’ until the ripe stage (12 days). For the varieties ‘Sierra Léone’ and ‘Sewe’, SSC were stable between 4 and 8 days of storage and increased by 19% and 42%, respectively, at 12 days of ripening storage. At this ripe stage, the SSC of the different varieties varied at an approximate average of 15.9 ± 1.1 °Bx with a maximum of 17.1 ± 1 °Bx (‘Sewe’). Over the entire duration of storage, ‘Boukodiekhal’ variety is particularly noteworthy, with a significant increase recorded every 4 days sampling. That SSC increase is mainly explained by the hydrolysis of starch into soluble sugars, such as sucrose, glucose and fructose [54,55]. Differences of SSC increase rates, which were higher between ‘Boukodiekhal’ and ‘Sewe’ and would be explained by a greater degradation of starch in these two varieties compared to the others.

The ripening process was accompanied by a decrease in TA of approximately 89–98% for all varieties. Excepted ‘Sewe’ mango, no significant changes were observed after the 8th day of storage. At this time of storage, similar rates of decrease in TA were observed on ‘Manila’ variety, at the same storage temperature (25 °C) [56]. Starting from contrasting levels at harvest, acidity levels declined rapidly to stabilize at the eighth day, at an approximate average of 0.11 g/100 g, with a maximum at 0.13 g/100 g, in ‘Boukodiekhal’ and ‘Sewe’ mangoes. At this ripe stage (12 days), ‘Diourou’, a less acidic variety (0.06 g/100 g), differs significantly from ‘Sierra Léone’ and ‘Papaye’, at 0.10 and 0.11 g/100 g, respectively. ‘Diourou’ and ‘Sewe’ presented, respectively, the maximum and minimum decrease rate of acidity. As observed on ‘Alphonso’ and ‘Banganapalli’ mangoes [27], different varieties studied showed contrasting rates of decay. This change in acidity is related to the fruit aerobic metabolism [53], involving mainly citric and malic acids [14], which are main organic acids in most mango varieties [57]. Conversely to TA, pH increased significantly during ripening, with minimum (2.81) and maximum (6.01) pH recorded on the same variety ‘Diourou’. Excepted the variety ‘Sierra Leone’ at pH 4.13, all mangoes are at pH < 4 until the fourth day of ripening. Overall, pH ranges are from [2.81–3.76] at the mature green stage to [3.09–4.13]; [4.34–5.43] and [4.75–6.01], respectively, at 4, 8 and 12 days of ripening storage. At the ripe stage (8 days), ‘Boukodiekhal’, ‘Papaye’ and ‘Sewe’ mangoes would present technological advantage, notably for fresh prepared food product, with a slightly firm mango to the touch but, above all, less perishable because they are less susceptible to undesirable biological and biochemical changes at pH < 4.6 [58].

Regarding pulp color, all local varieties had significant decrease of Hue angle from 4 days of storage, except for ‘Boukodiekhal’. With an inter-variety variability of only 5%, the Hue angle for all varieties combined, varied on average from 97.7 ± 2.6 to 81.0 ± 2.4 from the harvest stage to 8 days of ripening storage. From this stage, a significant decrease of the Hue angle led to the ripe stage (12 days), to pulps of contrasting colors comparing ‘Papaye’ and ‘Sierra Léone’ to ‘Sewe’ and ‘Boukodiekhal’, the ‘Diourou’ mango being on intermediate color (Hue) level. These results are similar to the general observation of a ripening mango pulp color, which varies from milky-green to yellow-orange, as described on five Thai varieties including ‘Nam Dokmai’ [59]. This results in a decrease of L* and an increase of a* and b* parameters as highlighted by Noiwan et al. [60] and then modeled by Penchaiya et al. [61]. The evolution of these colorimetric parameters, in this case the decrease of Hue, illustrates a pigmentation of the mango mesocarp from the endocarp to the epicarp [30]. During ripening, the chlorophyll degrades exposing the pigments already present as well as the bio-synthesized anthocyanins and carotenoids [62,63]. Conversely to the pulp Hue, carotenoid content increases during mango ripening [64,65].

After 8 days of storage, all of the mango varieties reached a fairly advanced level of ripeness. From the organoleptic point of view, an open skin, a yellow-orange turning pulp, but a fruit still rather firm by hand pressure, were perceptible. Comparing the different varieties with respect to the physicochemical properties (SSC, AT, pH, Hue) of the pulp, significant differences and some similarities were found during ripening (Figure 3). Several studies reported similar trends in the evolution of physicochemical characteristics of the pulp, suggesting a ‘variety’ effect. Moreover, physicochemical characterization of these varieties provides useful information for fruit fly control. In fact, within the same fruit species, the variety plays an important role on the oviposition preference of *Bactrocera invades* females [66].

### 3.4. Ripening Prediction

Mango pulp, whether consumed fresh or pureed, can be appreciated for its taste (sweetness, acidity), aroma and textural properties. By descriptive [67] and hedonic [68] methods, some studies have highlighted sensory varietal differences. Beyond the organoleptic aspect, these differences are reflected in physicochemical parameters, notably acidity (pH), sweetness (SSC) and color (Hue), which are accessible in rapid measurements. In this approach, quality represents the organoleptic profile of the mango, essentially defined by its ‘acid’ and ‘sweet’ flavors and the ‘yellow-orange’ color of pulp. Therefore, depending on the variety, it would be convenient to use these physicochemical parameters to predict the storage time corresponding to a given mango ripeness quality (acidity, sweetness, yellow-orange color).

#### 3.4.1. Intra-Varietal Predictions of Mango Ripening

From green-mature to ripeness, each physiological stage of mango corresponds to a range of physicochemical parameters. Comparing the physicochemical characteristics of ‘Alphonso’ and ‘Bangannapali’ [27], it appears that the same ripening stage (e.g., climacteric phase) can correspond different ripening time from one variety to another. Therefore, the storage time can validly serve as an indicator of mango ripening quality declined as a range [pH, SSC, Hue] for each variety.

Table 3 presents the parameters of ‘ripening storage time’ (R_ST_) predicting equations for each of the five local varieties by multi-linear regression. Overall, the R^2^, RMSE and MRD values, indicate a reliable prediction of R_ST_ response by pH, SSC and Hue explanatory variables. However, on ‘Boukodiekhal’, ‘Papaye’ and ‘Sewe’, RMSE and MRD values obtained in validation are relatively high. This can be explained by difference in ripeness level of samples due to the time lag between the harvest periods of the calibration and validation samples. This observation suggests an effect of harvest period on the prediction model. Thus, the decrease noted in RMSE, MRD and R^2^ of models using α-variables (Hue, pH, and SSC), confirm an efficiency by monitoring that physiological process, notably the Hue angle pulp affected by season advancement [69]. Furthermore, the levels of influence (*p* < 0.05) of the ‘models’ coefficients reveal varietal differences. It appears that in ‘Sierra Leone’ and ‘Sewe’, the pH variable does not significantly influence the R_ST_ response.

ANCOVA proved that in addition to the ‘variety’, the ‘ripening storage time’ (R_ST_), strongly correlated to TA, pH, SSC and Hue, was the most influential factor. Based on this premise, prediction of mango ripening will be done by determining the R_ST_ (ripening storage time) corresponding to a predefined quality as a range of physicochemical parameters. Thus, the parameters of rapid measurements: pH, SSC and Hue angle are the three physicochemical variables used to predict the ripeness quality of mango. These physicochemical parameters measured on the pulp have evolved significantly throughout the storage. This reflects the construction of the fruit quality during the ripening process inducing synchronous physicochemical changes.

For the example of the variety ‘Diourou’, the model (R_ST_ = 0.79 pH − 0.17 Hue + 0.55 SSC + 9.73) allows a prediction of the storage time, according to acidity, sweetness and pulp color indicators [pH, SSC, Hue]. This prediction model to [R^2^ = 0.99, MRD = 0.06] and [R^2^ = 0.89, MRD = 0.23], respectively, in calibration and cross-validation is a reliable model to determine the storage time (days) needed for ripe mangoes of a predefined ripeness quality [pH, SSC and Hue]. On the five models average, the R^2^ at 0.98 ± 0.01 (calibration) and 0.88 ± 0.05 (validation) show that these proposed models made good intra-varietal prediction.

#### 3.4.2. Multi-Varietal Prediction of Mango Ripening

Pearson’s test on the overall sample, approximately 100 mangoes from all varieties combined, reveals lower correlations (*p* < 0.05) between TA and SSC (R^2^ = 0.72) or Hue angle (R^2^ = 0.66), compared to intra-varietal correlation. This suggests the importance of a systemic approach involving variables characteristic of the ‘variety’ effect, which was confirmed (*p* < 0.05) by ANCOVA with pH, SSC and Hue variables. These quantitative variables are here the varietal parameters C_f_ and C_c_ (shape and size of the variety) or physiological C_tr_ (transpiratory water loss) determined beforehand.

Six prediction models are presented in Table 4, with Model 1, involving no varietal variables, serving as a control. The AIC is used to evaluate the quality of the statistical models in comparison. Model 2 shows good prediction of R_ST_ but no significant effect of C_tr_ in the model. With a lower AIC, model 3, involving the parameters C_f_ and C_c_, is the one guaranteeing the best compromise between goodness of fit and number of explanatory variables. R_ST_ with an R^2^ of 0.97 and the lower RMSE and MRD values of this model are thus related to a better prediction. Taken individually (model 5), the fruit shape parameter (C_f_), does not significantly impact the prediction of R_ST_. Looking at the models with only one varietal parameter (n° 4, 5 and 6), it appears that the ‘size’ factor C_c_ composes the best model (n° 4) with an AIC -40.5. This result suggests an important effect of the mango mass according to its size, on the ripening kinetic at storage. As an example, the colorimetric parameters, in this case the decrease of Hue, illustrates a pigmentation of the mesocarp of mango going from the endocarp to the epicarp [30].

The model n° 3 (R_ST_ = 0.99 pH − 0.14 Hue + 0.66 SSC − 0.09 C_f_ − 18.69 C_c_ + 16.33), having the lowest AIC, can be retained for a multi-varietal prediction in multi-linear regression.

The regression by adimensional variables (α) tested on the selected model (n° 3) results in an increase of the R^2^ and a decrease of the RMSE and MRD. The following model Formula (9), assessed by α-variables, thus allows a multivariate prediction of R_ST_ in MLR (AIC = −49.9) calibrated and validated, respectively, with MRD at 9% and 22%; and R^2^ 0.97 and 0.81.
R_ST_ (days) = 2.49 pH − 16.28 Hue + 4.70 SSC − 0.11 C_f_ + 4.06 C_c_ + 8.44(9)

pH, Hue and SSC coefficients are here expressed in relative values (α) according to Formula (2).

The α-variable model will also allow R_ST_ prediction, starting from a determined initial range (pH(0), SSC(0), Hue(0)), to reach a targeted range (pH(t), SSC(t), Hue(t)). A numerical application of the Formula (6) shows that from mangoes with an initial range (pH = 3, SSC = 9 °B, Hue = 90), it will take 9 days for ‘Boukodiekhal’, or 10 days for ‘Sierre léone’ mangoes to ripen at 25 °C/86%RH and reach the range (pH = 4.5, SSC = 18 °B, Hue = 70).

This study on these local varieties shows that with the help of statistical models, storage time can be predicted as an index of ripening stage. In a previous study, this parameter strongly correlated with physicochemical properties and it was used by Rosa María et al. [70] to compare consumer acceptability scores on freshly cut mangoes, depending on the ripening stage.

## 4. Conclusions

This study further positions these five local varieties in the local market and in the world varietal chessboard. Based on the varietal contrast in mango characteristics at mature stage and during storage, statistical models were proposed to predict the storage time required for a predefined quality [pH, SSC, Hue]. For an optimal exploitation of the mango in general and particularly of these under-exploited varieties, the intra-varietal prediction models and the multi-varietal models (specially the n° 3), will allow a better quality control of this perishable fruit. Moreover, the use of the adimensional value (α-variable form) in the proposed models, provides storage prediction for a ripening monitoring based on a known initial stage of mango. Furthermore, textural parameters or other associated to the harvest period in season could be included for improved models. This systemic approach should promote as part of decision support tools for the use of industrial processing.

## Figures and Tables

**Figure 1 foods-11-03759-f001:**
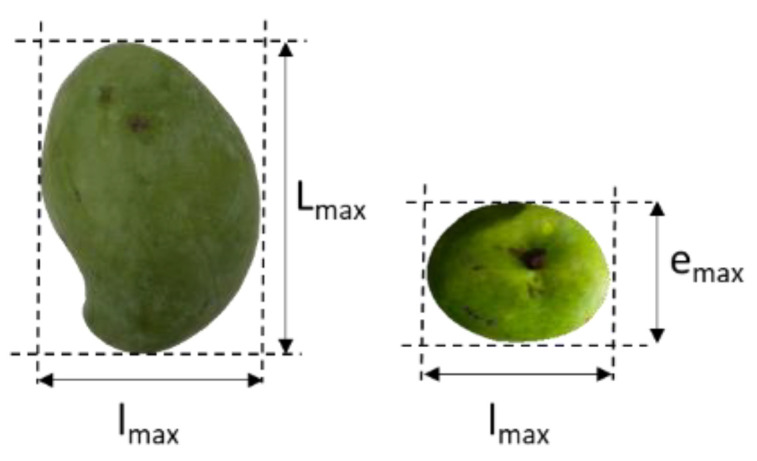
Digital photo of the three dimensions (L_max_, l_max_ and e_max_) of mango.

**Figure 2 foods-11-03759-f002:**
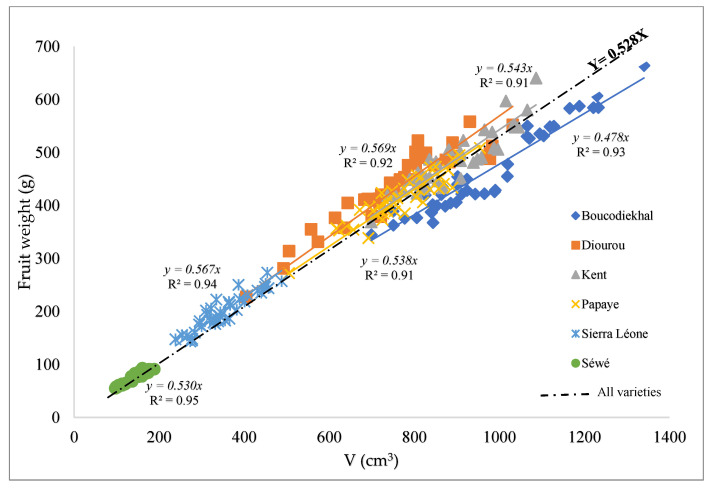
Correlation between the fruit weight and the equivalent volume V(cm^3^) = L_max_ × l_max_ × e_max_. The line Y = 0.528X (R^2^ = 0.96) is the linear regression on all observations of the scatterplot, all varieties combined (*n* = 240). The lines y = C_c_.x are the linear regressions per variety (*n* = 40).

**Figure 3 foods-11-03759-f003:**
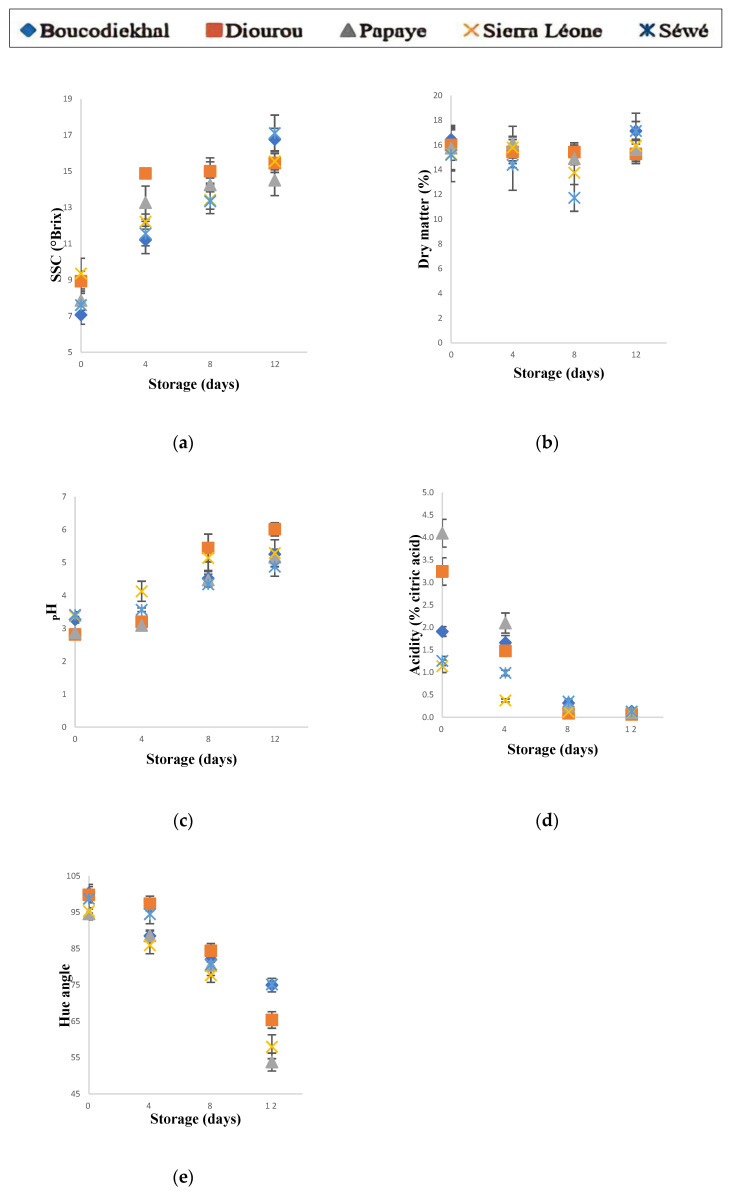
Physicochemical parameters of the five local varieties pulp during ripening storage. (**a**) SSC, (**b**) DM, (**c**) pH, (**d**) TA, (**e**) Hue. Each point represents mean value of *n* = 5 mango fruits × 2 trials per fruit with standard deviation bars.

**Table 1 foods-11-03759-t001:** Weight, dimension and form parameter (C_f_) values of different mango varieties.

Variety	Weight (g)	L_max_ (cm)	l_max_ (cm)	e_max_ (cm)	C_f_	Mango Form
Boukodiekhal	469.7 ^b^ (44.5)	12.6 ^a^ (0.9)	10.2 ^b^ (0.3)	7.6 ^b^ (0.5)	16.8 ^f^ (1.2)	elliptic to oblong
Diourou	442.2 ^bc^ (51.2)	11.2 ^c^ (0.6)	10.8 ^a^ (0.7)	8.2 ^a^ (0.7)	12.9 ^c^ (1.8)	elliptic
Papaye	423.3 ^c^ (50.7)	12.5 ^ab^ (0.9)	8.4 ^c^ (0.4)	7.2 ^b^ (0.3)	15.9 ^e^ (1.2)	oblong-reniform
Sierra Léone	214.4 ^d^ (31.5)	8.3 ^d^ (0.6)	8.4 ^c^ (0.5)	6.2 ^c^ (1.0)	11.1 ^b^ (0.8)	elliptic-reniform
Sewe	102.5 ^e^ (23.4)	5.1 ^e^ (0.5)	6.7 ^d^ (0.5)	5.2 ^d^ (0.4)	6.7 ^a^ (0.8)	oblong
Kent *	600.6 ^a^ (69.2)	12.0 ^b^ (0.9)	10.0 ^b^ (0.7)	8.5 ^a^ (0.6)	13.8 ^d^ (1.6)	oblong-roundish

Values in brackets represent the standard deviation on respective mean values (*n* = 40 fruits). Superscript letters within the same column indicate significant differences between varieties (*p* < 0.05); * Reference variety.

**Table 2 foods-11-03759-t002:** Percentage of weight lost at 12 days (ripe stage), loss rate (slope in %/day) and linear regression coefficient (R^2^) as a function of variety and storage conditions (AC: ambient; CC: controlled 25°C/86% RH).

Variety	Weight Loss (%) at Ripe Stage (12 Days Storage)		^1^ Loss Rate (% Per Day)		R²	
	CA	CC	CA	^2^ CC(C_tr_)	CA	CC
Boukodiekhal	27.25 ^a^ (2.05)	12.19 ^a^(1.41)	2.35	1.05	0.96	0.98
Diourou	16.72 ^b^ (1.24)	9.88 ^ab^ (1.53)	1.35	0.76	0.97	0.99
Papaye	13.20 ^bc^ (1.72)	6.70 ^cd^ (0.99)	1.07	0.52	0.98	0.98
Sierra Léone	25.52 ^a^ (2.12)	8.73 ^bc^ (0.75)	2.22	0.75	0.98	0.98
Sewe	22.99 ^a^ (2.91)	9.70 ^abc^ (2.19)	1.93	0.85	0.98	0.98
Kent *	10.56 ^c^ (1.15)	4.71 ^d^ (0.42)	0.93	0.41	0.94	0.98

Superscript letters in the same column indicate significant inter-variety differences (Tukey test, *p* < 0.05). Values in brackets represent the standard deviation on respective mean values (*n* = 5 mango fruits) ^1^ Loss rate represents the directing coefficient (slope a) of linear equation: ‘Weight loss (%) = a × t (days)’. ^2^ Loss rate values in CC storage stand for the varietal parameter (C_tr_). * Reference variety.

**Table 3 foods-11-03759-t003:** Model coefficients and parameters for intra-varietal prediction of mango ripening by multi-linear regression.

Variety	^1^ Variable Form	Coefficients of Variables				R^2^		RMSE		MRD	
		Intercept	pH	Hue	SSC	Calibration	Validation	Calibration	Validation	Calibration	Validation
Boukodiekhal	X	5.148	1.346 *	−0.133 *	0.554 **	0.975	0.838	0.798	4.701	0.097	0.847
	α	5.148	4.393 **	−13.377 *	3.916 **	0.975	0.853	0.798	2.427	0.097	0.385
Diourou	X	9.729	0.79 **	−0.168 ***	0.548 ***	0.994	0.891	0.402	2.036	0.056	0.223
	α	9.729	2.221 **	−16.810 ***	4.890 ***	0.994	0.891	0.402	2.040	0.056	0.226
Papaye	X	1.277	1.75 **	−0.109 ***	0.523 ***	0.988	0.814	0.559	3.771	0.097	0.533
	α	1.277	5.031 **	−10.273 ***	4.108 ***	0.988	0.817	0.559	2.911	0.097	0.276
Sierra Léone	X	1.442	0.998 ns	−0.13 **	0.821 **	0.965	0.964	0.941	1.680	0.133	0.147
	α	1.442	3.368 ns	−12.375 **	7.673 **	0.965	0.964	0.941	2.031	0.133	0.192
Swe	X	4.285	1.367 ns	−0.14 **	0.679 ***	0.988	0.881	0.555	3.452	0.079	0.524
	α	4.285	4.665 ns	−13.857 **	5.166 ***	0.988	0.888	0.555	2.013	0.079	0.235

^1^ X: variable parameters expressed in absolute values. ^1^ α: variable parameters expressed in relative values (adimensional form). * *p* < 0.05, ** *p* < 0.01, *** *p* < 0.0001, ns = no significative.

**Table 4 foods-11-03759-t004:** Model coefficients and parameters for multi-varietal prediction of mango ripening by multi-linear regression.

Models	Coefficients of Variables							AIC	R²		RMSE		MRD	
	Intercept	pH	Hue	SSC	C_tr_	C_f_	C_c_		Calibration	Validation	Calibration	Validation	Calibration	Validation
1 (Control)	4.440	0.996 ***	−0.132 ***	0.665 ***				−9.6	0.958	0.748	0.935	3.093	0.143	0.446
2	17.817	1.060 ***	−0.134 ***	0.662 ***	−0.731 ns	−0.100 **	−21.614 ***	−52.9	0.974	0.704	0.730	3.366	0.102	0.478
3	16.335	0.989 ***	−0.139 ***	0.660 ***		−0.090 **	−18.698 ***	−53.8	0.974	0.705	0.734	3.344	0.102	0.476
4	12.038	1.090 ***	−0.131 ***	0.657 ***			−14.836 ***	−40.5	0.970	0.704	0.793	3.357	0.117	0.477
5	5.000	0.966 ***	−0.134 ***	0.666 ***		−0.022 ns		−8.3	0.958	0.751	0.931	3.072	0.141	0.443
6	5.302	0.781 ***	−0.152 ***	0.655 ***	2.386 ***			−26.1	0.965	0.728	0.852	3.161	0.125	0.453

** *p* < 0.01, *** *p* < 0.0001, ns = no significative.

## Data Availability

The data presented in this study are available on request from the corresponding author.

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
