# Peer review of "Storage Time as an Index for Varietal Prediction of Mango Ripening: A Systemic Approach Validated on Five Senegalese Varieties"

_foods, 2022, doi:10.3390/foods11233759_

Round 1

Reviewer 1 Report

Storage Time as Index for Varietal Prediction of Mango Ripening: a systemic approach validated on five Senegalese Varieties

The main purpose of this work was to study prediction models integrating variables deduced from varietal characteristics for five mango varieties

This work is well written; clearly and objectively, however, I would like to suggest some questions/comments describe below.

Page 3, line 87:

The correct form is “ripening index”

Pages 4 and 5, Statistical analyses

Normality tests usually have as the null hypothesis the normality of the data.

Lines 185-187 is shown that "After verification of normal distribution of the residuals (p<0.05), the multi-linear regressions were tested for the storage time prediction."

Therefore, for most normality tests, p-value < 0.05 indicates that the data do not follow the Normal distribution.

In this case, the proposed methodology is not indicated.

What was the normality test used and the p-values obtained in this test? Please clarify.

Page 10, lines 351-353:

Change this sentence for: “Changes in soluble solid content (SSC), dry matter (DM), pH, titratable acidity (TA) and color (Hue) pulp during ripening can be observed respectively in Figure 4.(a).; 4.(b).; 4.(c).; 4.(d). and 4.(e).”

Or

Change the order of the figures so you don't need to change all the text.

Page 12, lines 418-419:

The same observation is considered about the orders of the figures.

Page 15, line 528:

The correct is “according to formula (9)”.

Besides, it is necessary to carry out a citation review of all formulas in the manuscript completely.

Reviewer 2 Report

This study aimed to predict the storage time of five Mango during ripening, but the experimental design is too simplistic and some indexes missed. For example, the appearance changes of Mango during the storage time, the changes of respiratory rate of Mango and so on. In addition, there are some small mistakes in sentence expression in the manuscript. 

Reviewer 3 Report

Mango fruits are not only one of the most consumed in the world, but their production is an important part of the agricultural economy of many countries with tropical and subtropical climates. The paper ”Storage Time as Index for Varietal Prediction of Mango Ripening: a systemic approach validated on five Senegalese Varieties” is devoted to the description of the fruits of local varieties, which, as the Authors described, are poorly characterized so far. The paper also presents the most important factors that were used to build predictive models of mango ripening. The Authors' metodological approach may be of help to researchers studying other climacteric fruits. This is a great advantage of this article, while the results themselves have local significance. It is a pity that the Authors did not supplement their research with an analysis of consumer preferences as to the stage of fruit ripeness of the studied varieties. Such a simple test would help to optimize the fruit storage process and would certainly be of economic importance.

In the Introduction, there is no information about the origin of local varieties (it is mentioned that they are endemic only in the Abstract). For this reason, when reading the Introduction, it is not known if they come from Senegal or if they were brought and domesticated long ago, and therefore perhaps grow elsewhere.

The Results and Discussion chapter is too long. The Authors have described in detail what the reader can find in the Tables. There is no need to report the results in two forms. By shortening this chapter, the discussion itself will become clearer.

Lengthy conclusions discourage reading. They are too long. The sentences describing the experiment (”The characteristics of mangoes harvested at the green-mature stage were determined. Weight losses of mango fruit as well as physicochemical parameters of the pulp, were monitored over 12 days of ripening storage.”) peculiar to the Abstract, are unnecessary in Conclusions.

The Figures are very carefully made, but I have a minor objection to them. I have printed the manuscript in grayscale and I suggest the Authors to do it as well. In this version, the markers used in Figures 2, 3 and 4 are partially indistinguishable. It would be worth giving the markers some additional features, e.g. different shape, frame, darkening or lightening (such a procedure would also help people who have problems with distinguishing colors).

I would like to draw the Authors' attention to a number of inconsistencies in spelling. Note the spelling of the various parameters in the equations. For example, RST is written in simple letters in some equations (6 and 8) and in italics (eq. 4 and 5). Basically, in science textbooks they should be written in italics (except for Greek symbols). In general, I do not quite understand why some equations are written straight and others in italics.

The spelling of the variety names should be the same throughout the text, and here we have four versions: "kent" (page 1, line 19), 'Kent' (p. 1, l. 43), 'kent' (p. 9, l. 31), and Kent (p. 10, l. 37).

There are a lot of typos in the text. In some places the Authors used double spaces which left gaps (eg. page 1. line 38; page 2 line 64). Many spaces are missing elsewhere in the manuscript. The Authors should pay attention to writing a space before and after the equal sign or the sign "less than" (the notation is inconsistent already in the Abstract), and to placing a space before the gram symbol abbreviation.

There is a problem with multiplication sign in the text. It should be a symbol from the character table, but not the letter x. Equations with the multiplication sign are written in two ways: with spaces (page 6, line 22, page 7 line 28) or without spaces (last paragraph on page 7).

The phrase "3D dimension" (page 2, line 68) is incorrect because the abbreviation 3D comes from the word dimension/dimensional.

I would write ”Senegalese” with an uppercase letter in keywords.

There is no empty line after the description of Table 2.

Concluding, the entire manuscript should be read very carefully, some parts should be shortened and any minor errors and shortcomings should be eliminated.

Oct. 25th, 2022

Reviewer 4 Report

RESULTS AND DISCUSSION section

Line 250: Change the order 'Boukodiekhal' and 'Papaye'.

Line 257: Delete 'Sierra Leone' (coefficient of variation for fruit weight of this variety is 14%)

Line 260: Put variety instead of lot

Line 350: 3.3. Physicochemical parameters of mango pulp at harvest and ripening storage

Considering that in the previous tables (Tab. 1 and 2) and graphs (Figures 2 and 3) you showed the parameters of mango quality for the variety 'Kent', which represents the reference variety, it would be desirable that in the graphs 4.(a) .; 4.(b).; 4.(c).; 4. (d). and 4.(e) you should put the change of the tested parameters during ripening storage for this reference variety in addition to the tested local varieties.

Line 477: Table 3. Add data for the cultivar 'Kent' too (reference cultivar)

ADDITIONAL COMMENTS

The aim of the paper entitled „Storage time as index for varietal prediction of mango ripening: a systemic approach validated on five Senegalese varieties“, according to the authors, was to determine for five local mango cultivars, based on monitoring weight loss and physical and chemical parameters during 12 days of storage, statistical models for predicting the storage time required for a predefined mango quality. The proposed intravarietal prediction models and multivarietal models according to the authors will enable better quality control of this perishable fruit.

In my opinion, the objectives of this work are interesting. The introduction is interesting, it explains very well the problems related to mango storage. In the Materials and Methods section, the methods are explained in detail. However, in the Results and Discussion section, in the presented tables and graphics, there’s a lack of data related to the cultivar 'Kent' which the authors took as a reference cultivar, and in my opinion it is necessary to add that data as well.

Round 2

Reviewer 2 Report

I have no comments.